# Differential Dynamics and Roles of FKBP51 Isoforms and Their Implications for Targeted Therapies

**DOI:** 10.3390/ijms252212318

**Published:** 2024-11-16

**Authors:** Silvia Martinelli, Kathrin Hafner, Maik Koedel, Janine Knauer-Arloth, Nils C. Gassen, Elisabeth B. Binder

**Affiliations:** 1Department Genes and Environment, Max Planck Institute of Psychiatry, Kraepelinstr. 2-10, 80804 Munich, Germany; 2Institute of Computational Biology, Helmholtz Munich, Ingolstaedter Landstraße 1, 85764 Neuherberg, Germany; 3Research Group Neurohomeostasis, Department of Psychiatry and Psychotherapy, University of Bonn, Venusberg Campus 1, 53127 Bonn, Germany

**Keywords:** FKBP5, stress, isoforms

## Abstract

The expression of *FKBP5*, and its resulting protein FKBP51, is strongly induced by glucocorticoids. Numerous studies have explored their involvement in a plethora of cellular processes and diseases. There is, however, a lack of knowledge on the role of the different RNA splicing variants and the two protein isoforms, one missing functional C-terminal motifs. In this study, we use in vitro models (HeLa and Jurkat cells) as well as peripheral blood cells of a human cohort (*N* = 26 male healthy controls) to show that the two expressed variants are both dynamically upregulated following dexamethasone, with significantly earlier increases (starting 1–2 h after stimulation) in the short isoform both in vitro and in vivo. Protein degradation assays in vitro showed a reduced half-life (4 h vs. 8 h) of the shorter isoform. Only the shorter isoform showed a subnuclear cellular localization. The two isoforms also differed in their effects on known downstream cellular pathways, including glucocorticoid receptor function, macroautophagy, immune activation, and DNA methylation regulation. The results shed light on the difference between the two variants and highlight the importance of differential analyses in future studies with implications for targeted drug design.

## 1. Introduction

The FK506 binding protein 51 (FKBP51) is a ubiquitously expressed immunophilin, encoded by the gene *FKBP5*, and whose function has been investigated in association with numerous biological processes describing FKBP51 as a central regulator of pathways involved in psychiatric and neurodegenerative disorders, immune response, inflammation, cardiovascular diseases, metabolic pathways, and cancer [1,2,3,4,5]. The most investigated role of FKBP51 is its involvement in the regulation of the stress response, initially discovered in squirrel monkeys, where it was observed that an increased expression of FKBP51 is the cause of glucocorticoid receptor (GR) resistance and high circulating cortisol levels in these animals [6,7]. The critical role of FKBP51 in hypothalamus–pituitary–adrenal (HPA) axis regulation is also supported by the fact that feedback control of this axis is impaired in FKBP51-deficient mice. In fact, FKBP51 is an inhibitor of the GR, the key effector of the hypothalamus–pituitary–adrenal (HPA) axis. On the other hand, it is also a transcription target of the GR, with several glucocorticoid response elements (GREs) in intronic and upstream enhancer regions and strong upregulation observed across many tissues. This can lead to an ultra-short negative feedback of GR activity [8]. FKBP51 is a co-chaperone protein and interacts not only with the GR via HSP90 but also with many other proteins via direct protein–protein interactions [9,10]. These interaction partners include heat shock proteins, steroids receptors, PH domain and leucine-rich repeat protein phosphatases (PHLPP) and Akt, Nuclear Factor ‘Kappa-Light-Chain-Enhancer’ of Activated B-Cells (NF-κB) as well as DNA methyltransferase 1, Calcineurin-NFAT signaling, Tau, and others [11]. These interactions have been shown to play a relevant role in many cell types, including cancer cells [11,12]. Given the strong upregulation of FKBP51 following glucocorticoid exposure and its many downstream partners and thus central role in promoting a cellular stress response, it is a particularly interesting potential drug target for a number of diseases.

In humans, four transcription variants (variants 1–4) of FKBP5 have been annotated, coding for two different isoforms (isoform 1 and 2) of the protein FKBP51. In mice, a species from which a large body of current knowledge on FKBP5/51 is derived, only one isoform (corresponding to the human isoform 1) of FKBP51 is annotated. The human transcription variant 1 (ENST00000357266.8) differs in the 5′ UTR from variants 2 (ENST00000536438.5) and 3 (ENST00000539068.5) and all three code for the 475 amino acid (aa) long isoform 1 (Q13451-1), while variant 4 (ENST00000542713.1) corresponds to a much shorter transcript and codes for the “truncated” isoform 2 (Q13451-2) of 268 aa (Figure 1a,b). Gene expression data (gtexportal.org, accessed on 14 September 2023; [13]) indicate that FKBP5 is ubiquitously expressed with particularly high expression levels in the tibial nerve, skeletal muscle, and esophagus (Figure 1c). Variant 1 is more strongly expressed than the other variants, suggesting that in all experiments conducted without distinguishing between the different variants, the overall expression levels mirror mainly the ones of variant 1. Variant 2 appears to be the least expressed, while there does not seem to be a quantitative difference between variants 3 and 4 but rather a tissue-specific differential expression.

At the protein level, FKBP51 isoform 1 has two N-terminal FK506-binding (FK) domains and three tetratricopeptide repeat (TPR) motifs at the C-terminus (Figure 1b). Isoform 2 of FKBP51 shows sequence identity with isoform 1 for the first 222 amino acids (aa), corresponding to the FK domains. The sequence ranging from aa 223 to its C-terminus (aa 268) is unique and, so far, uncharacterized. Missing the rest of isoform 1’s C-terminal region, isoform 2 therefore lacks the TPR motifs (Figure 1b). The first FK domain, FK 1, is the binding site of the immunosuppressive drug FK506, from which the protein gets its name. FK1 also exerts a peptidyl-prolyl cis-trans isomerase (PPIase) or rotamase activity [14], characteristic of all immunophilins. The pocket in FK1 is also the binding site for another drug, rapamycin. This drug, in a complex with FKBP51, exerts immunosuppressive and anticancer effects, mediated via the selective inhibition of the mechanistic target of rapamycin or mTOR [15]. Downstream, adjacent to FK1, is the second FK domain, FK2, which is presumably derived from a duplication event of the FK1 domain and shares 32% sequence homology with it [16] but lacks measurable rotamase activity [17] and does not bind FK506. Instead, it might have cooperative functions with the TPR motifs [17]. The TPR motifs at the C-terminus promote protein–protein interactions [18], in particular with chaperone proteins such as HSP90 and heat shock protein 70 (HSP70) [19]. Furthermore, Li and colleagues showed that the TPR motifs are also responsible for the interaction with the serine–threonine phosphatase calcineurin (CaN) [20]. This phosphatase activates nuclear transcription factors of activated T lymphocytes (NFAT), responsible for the expression of interleukin-2 (IL2) and several T cell-specific activators, thereby regulating the clonal expansion of T cells after stimulation by an antigen [20]. Thus far, only a few studies carried out by the group of M.F. Romano at the University of Naples, Italy, have described different functions of the human isoforms, with isoform 2 associated with the development of melanoma and glioma [21,22]. Given the substantial structural difference between the two FKBP51 isoforms and the scarcity of studies regarding their differential roles, we decided to investigate possible differential functions of FKBP51 isoforms. We first mapped transcript and isoform differences in the expression dynamics following induction by glucocorticoids and then probed functional differences in key pathways.

**Figure 1 ijms-25-12318-f001:**
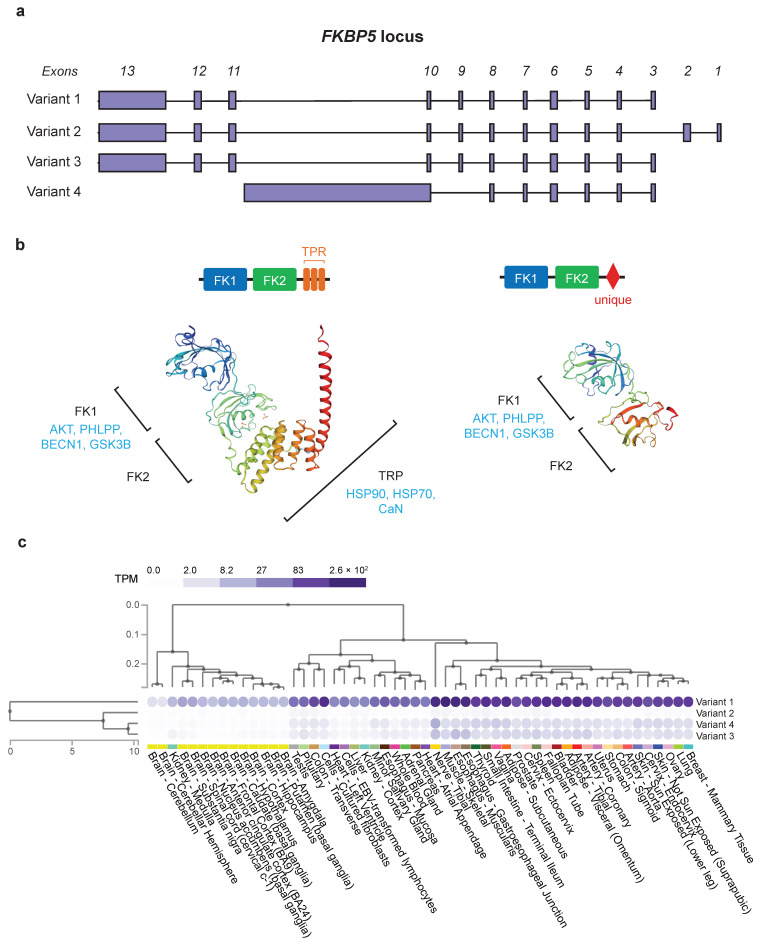
FKBP5/51 transcription variants and isoforms: (**a**) Schematic view of the *FKBP5* locus on human chromosome 6 and the four splicing variants of the gene (adapted from gtexportal.org). (**b**) Schematic view of FKBP51 isoform 1 and 2 protein structures and 3D structure models generated with the Swiss model repository server of the expasy portal (swissmodel.expasy.org (accessed on 16 October 2024); [23]). Domains are indicated in black and experimentally validated domain-associated binding partners in blue. (**c**) Transcription variant-specific *FKBP5* expression throughout human tissues (adapted from gtexportal.org; [13]). The data used for the analyses described in this figure were obtained from: www.gtexportal.org, the GTEx Portal on 14 September 2023. The terms and conditions for the use of data and images can be found here: https://www.gtexportal.org/home/downloads/adult-gtex/overview, accessed on 14 September 2023.

FKBP51 is strongly activated with HPA-axis activation, and altered HPA-axis function has been observed with many disorders, including psychiatric, cardiovascular, and autoimmune diseases [24]. FKBP51 is also strongly upregulated with synthetic glucocorticoids, and these are commonly used in the treatment of many conditions and disorders [25]. Understanding differences in the regulation of FKBP51 isoforms following glucocorticoid exposure could have important clinical implications, especially if altered downstream pathways are different. In addition, FKBP51 is actively explored as a drug target in a number of diseases, including posttraumatic stress disorder, neurological diseases, pain, inflammatory disease, and cancers [26,27,28,29,30]. A better understanding of the function of different isoforms is important to optimize the development of FKBP51-targeting drugs.

## 2. Results

### 2.1. Expression and Degradation Dynamics of FKBP5/FKBP51

In order to characterize the expression dynamics of FKBP5, we first determined the expression via reverse transcription–quantitative polymerase chain reaction (RT-qPCR) in HeLa cells. Results evidence the absence of significant expression of variants 2 and 3. The probes covering all FKBP5 variants yielded the strongest signal, while variant 4 showed lower yet measurable levels (Figure 2a and Appendix A). This indicates that a large part of this signal derives from variant 1 expression. Given the key role of FKBP5/51 in the stress response, we were interested in the differential expression dynamics of the transcription variants upon GR activation. For this purpose, we stimulated HeLa cells with 100 nM of the GR agonist dexamethasone (Dex) for 2, 4, 6, 12, and 24 h. Transcription levels of the different mRNA variants were subsequently analyzed via RT-qPCR (Figure 2b). Due to the lack of sequence uniqueness for variant 1, probes spanning variants 1, 2, and 3 were used. Considering the absence of variants 2 and 3, the observed signal was assumed to correspond to variant 1 and will be referred to as variant 1 from here on. The expression of both variants 1 and 4 was significantly increased in response to Dex across time (Two-way ANOVA, time effect *p* < 0.0001) and a significant difference between variants over time (two-way ANOVA, time × variant effect *p* < 0.0005). Interestingly, despite having lower expression at basal levels, variant 4 showed an increased response ratio over vehicle compared to variant 1 at early time points (significant difference at 2 and 4 h). Furthermore, variant 4 showed a more rapid response to Dex than variant 1: variant 4 levels were significantly increased already after two hours of treatment, while at the same time point, variant 1 was still expressed at baseline levels. After 6 h treatment and until the end of the treatment period at 24 h, both variants showed a significantly increased expression compared to baseline but with no difference between each other: variant 4 follows a steady slope after 6 h treatment, while variant 1 expression reflects in a slowly increasing curve up to 24 h.

Having seen a difference in the response to Dex between variant 1 and variant 4 in cell culture, we decided to assess whether this finding holds true in a different tissue in vivo. For this purpose, we analyzed, via RT-qPCR, the expression of FKBP5 variant 1 and variant 4 in peripheral blood samples of a cohort of 26 healthy male participants at baseline and 1, 3, 6, and 23 h after 1.5 mg Dex intake. The results show a much lower expression of variant 4 compared with variant 1, as can be appreciated from the Ct values (Appendix A), which is in accordance with data for peripheral blood from GTEX (gtexportal.org; [13]). Upon Dex stimulation, we observed an increased expression in both variants but with a significantly different expression dynamic over the time course (Figure 2c; two-way ANOVA time effect *p* < 0.0001 and time × variants effect *p* < 0.0001). While a significant increase was detected for both variants after 3 and 6 h of Dex induction compared with baseline, variant 4 showed an increased expression already after one hour in contrast to variant 1, confirming the faster dynamic observed in vitro (Figure 2c; two-way ANOVA variants effect *p* < 0.0058). Interestingly, in blood samples, we can observe a faster dynamic compared to cell culture material, with a return to baseline levels after 23 h.

After having observed a different expression dynamic of variants 1 and 4 over time in response to Dex, we investigated the half-life of the respective protein isoforms: isoforms 1 and 2. For this purpose, a pulse-chase approach was used. HeLa cells were transfected with HaloTag^®^-tagged plasmids coding for either isoform 1 or 2. Twenty-four hours later, cells were tagged with a cell-permeable halogenated fluorophore 16, 8, 4, and 2 h before harvesting. After harvesting the cells, proteins were extracted and subjected to Western blot, and fluorescence intensity was measured on the nitrocellulose membrane. Results indicate that both isoforms are degraded throughout the 24 h (two-way ANOVA time effect *p* = 0.0001) at a significantly different rate (two-way ANOVA isoform effect *p* < 0.0001 and time × isoform effect *p* < 0.0001). The degradation of isoform 2 is faster, with a half-life of four hours, while isoform 1 reached 50% degradation only after 8 h (Figure 2d).

These results suggest a faster turnover of variant 4/isoform 2 with an increased and faster responsiveness to Dex and a shorter half-life of the protein compared with variant 1/isoform 1.

### 2.2. Differential Localization and Regulation of Cellular Pathways

To better understand possible differences in their functions, we analyzed the intracellular localization of the two isoforms. Overall, the information about FKBP51’s intracellular localization appears to be highly dependent on antibodies used for detection, and no information is available for the different isoforms. To avoid potential artifacts deriving from immunocytochemical processing, HeLa cells were transfected with plasmids coding for GFP-tagged isoforms 1 or 2. A plasmid expressing only GFP was used as control. Cells were live imaged 24 h after transfection. Resulting images (Figure 3a) showed ubiquitous signals from the control-transfected cells. Isoform 1 presented a cytoplasmic accumulation, while isoform 2 showed a distinct subnuclear localization. In support of this result, a motif analysis performed with the Expasy Prosite database (prosite.expasy.org, accessed on 14 September 2023) revealed a possible bipartite nuclear localization signal (NLS) between aa 232 and 246 (Appendix A), which corresponds to the region of the protein that is unique for isoforms 2 compared with isoform 1. In fact, despite having a low confidence level (score 3.000), the same analysis performed on isoform 1 could not detect any NLS (Appendix A).

Given the structural and subcellular localization differences, we investigated whether these have a functional effect. To this aim, we investigated different cellular pathways that are known to be regulated by FKBP51 and analyzed the differential role of the two isoforms on them.

As one of the best-known functions, we first analyzed the negative regulation of the two isoforms on GR. The activity of the different isoforms on GR was assessed via Glucocorticoid Response Element (GRE)-driven reporter gene assays. HeLa cells were co-transfected with MMTV-Luc, a GRE-driven luciferase, and with a plasmid coding for either isoform 1, isoform 2, or an empty vector as a control (ctr vector). Cells were then treated with increasing concentrations of Dex, and luminescence was measured 48 h after transfection (Figure 3b). As expected, GRE activity was enhanced in proportion to Dex concentration in the presence of both isoforms and the control vector (two-way ANOVA Dex effect *p* < 0.0001). Cells overexpressing isoform 1 showed a significantly lower dose–response curve compared with cells overexpressing either isoform 2 or the control plasmid (two-way ANOVA isoform effect *p* < 0.0001 and isoform × Dex effect *p* < 0.0001), which, in turn, were perfectly overlapping. Isoform 1 reduces the activity of GR, meaning that higher concentrations of Dex are required to evoke GR activation. To confirm these findings, the reporter gene assay was repeated with FKBP51 knock out (KO)cells. A CRISPR-Cas 9 approach followed by clonal selection was used to generate cells lacking isoform 1 only (iso 1 KO) or both isoforms (full KO) in HeLa cells using a pool of different guide RNAs (see Section 4 and Appendix A). With all genotypes, we saw, as expected, a dose-dependent curve in response to Dex (two-way ANOVA Dex effect *p* < 0.0001). The curve resulting from the luciferase assay in the full KO overlapped with the one from isoform 1-KO (i.e., still containing isoform 2). Both KO lines showed an overall increased activity with lower Dex doses as compared with WT (two-way ANOVA isoforms effect *p* = 0.0008). This result suggests that the lack of isoform 1 increases the sensitivity of GR to Dex and that isoform 2 alone (as seen in isoform 1 KO) is not able to rescue this effect (Figure 3c). Taken together, the results of both reporter gene assays indicate that isoform 1 alone, and not isoform 2, has an inhibitory function on GR.

Next, we proceeded with the analysis of the main macroautophagy markers, since it has been shown that this pathway is regulated by FKBP51 [31]. Upstream regulation of autophagy is tightly controlled by the kinase AKT. AKT (activated when phosphorylated) inactivates the autophagy initiator BECN1 via phosphorylation. In turn, AKT can be inactivated through dephosphorylation by the phosphatase PHLPP. This latter process is mediated by FKBP51 [31]. Isoform 1, 2, or an empty vector as control were overexpressed in HeLa cells, and the key markers of macroautophagy were analyzed via Western blot. Quantifications of pAKT showed that overexpression of both isoforms 1 and 2 led to a decreased phosphorylation of AKT (pAKT) compared with control (Figure 3d,e). Decreased pAKT leads to enhanced autophagy; therefore, we analyzed the main autophagic markers: BECN1, upstream regulator of autophagy, which is modulated directly by AKT; ATG12, involved in the expansion of autophagosomes being covalently bound to ATG5 and targeted to autophagosome vesicles (ATG12-ATG5); and LC3B-II (lipidated form of LC3B-I), marker of autolysosome formation. Overexpression of isoform 1 led to an increase in BECN1 and ATG12-ATG5 (Figure 3d,f,g). Interestingly, overexpression of isoform 1 did not lead to an increase in LC3B-II (normalized on LC3B-I) (Figure 3d,h). Furthermore, overexpression of isoform 2 did not affect levels of BECN1 but led to increased ATG12-ATG5 and LC3B-II/I (Figure 3d,f–h).

As we have previously shown, FKBP51 modulates DNA methyltransferase 1 (DNMT1) activity via phosphorylation in response to antidepressants, affecting genome-wide methylation levels [32]. To test the effect of the two FKBP51 isoforms on the phosphorylation (i.e., activation) levels of DNMT1 (pDNMT1), isoforms 1 or 2 of FKBP51 were again overexpressed in HeLa cells. pDNMT1 was detected via Western blot analysis and normalized to total DNMT1. Quantifications indicated a large reduction in pDNMT1 in the presence of isoform 1 overexpression (Figure 3d,i). Contrarily, overexpression of isoform 2 did not affect DNMT1 phosphorylation compared to control (Figure 3d,i).

FKBP51 has also been shown to be involved in the regulation of the immune response through Calcineurin-NFAT signaling [20]. We analyzed the effect of FKBP51 isoform 1 and 2 overexpression on the phosphorylation of NFAT. Given the importance of a proper immune response, we used the immortalized human T lymphocyte cell line Jurkat for this purpose. Plasmids coding for isoforms 1 or 2 of FKBP51 were overexpressed in Jurkat cells, and pNFAT levels were analyzed via Western blot. Quantifications revealed an increase in pNFAT when overexpressing isoform 1 (Figure 3d,j). Conversely, overexpression of isoform 2 did not affect pNFAT levels compared with control (Figure 3d,j).

Overall, these data revealed that the two FKBP51 isoforms can have equivalent or opposite effects. The reasons behind this and the possible implications will be examined in the Discussion.

## 3. Discussion

With this study, we highlighted both commonalities, as well as fundamental differences, between the two isoforms of the human FKBP51 protein, which could have implications for the design of isoform-selective compounds. Using targeted assays, we were able to map differences in Dex responsiveness and half-life of the two isoforms. In fact, in cultured cells, as well as in human blood samples, the short variant 4/isoform 2 appears to have a faster turnover, with a more rapid increase in expression upon dexamethasone treatment and a faster protein degradation rate in vitro. The faster dynamic that we observe in blood samples compared to HeLa cells, with a return to baseline levels after 23 h of Dex intake, is most probably due to a metabolization of Dex that occurs in vivo but not in vitro [33]. The general faster responsiveness of variant 4 mirrors the findings of Marrone and colleagues [34], albeit in a different context, where they observe a similar rapid response of variant 4 when stimulating T cell proliferation as compared with variant 1. Notably, the authors also report a nuclear localization of variant 4, which aligns with our own observations (Figure 3a). In silico analyses performed with the Expasy Prosite database (https://prosite.expasy.org/) revealed the presence of a putative nuclear localization signal (NLS) inside the unique C-terminal sequence of isoform 2 (Appendix A), validating the hypothesis of a selective nuclear localization and function of isoform 2 compared with isoform 1. Collectively, these corroborative findings suggest a potentially unexplored and important role for variant 4 in immediate-response transcriptional processes, warranting further investigation.

On a functional level, our experiments also revealed a partially distinct role for the two isoforms in the different cellular pathways. The two isoforms were found to exert distinct regulatory effects on GR, NFAT, and DNMT1 signaling. The regulation of these pathways depends on the interaction of FKBP51 with HSP90. It is therefore not surprising that isoform 1 affects the inhibition of GR and phosphorylation of NFAT and DNMT1, while isoform 2 does not have any effect on their function since it lacks the TPR domain responsible for the interaction with HSP90. Cells expressing isoform 2 alone are therefore not able to evoke the same GR response as WT, nor the same dephosphorylation and phosphorylation levels of DNMT1 and NFAT, respectively. On the other hand, the autophagic pathway, which is regulated via the interaction of FKBP51 with AKT and PHLPP, is modulated by both isoforms, since the interaction is dependent on the FK1 domain. The immediate consequence of FKBP51‘s interaction with AKT and PHLPP is the dephosphorylation of AKT. Interestingly, while AKT dephosphorylation is equally regulated by the two isoforms (Figure 3a), downstream effects, such as an increase in autophagy markers, are not (Figure 3e,f). This finding suggests the existence of an additional mechanism for which isoform 2 has a decreased effect on autophagy activation. Presumably, isoform 2 has a lower binding affinity for BECN1. Interestingly, though, isoform 2 appears to have a stronger effect in later stages of the autophagic pathway (autophagosome expansion and autolysosome formation), suggesting an alternative pathway or a faster activity of isoform 2 compared with isoform 1. The data suggest that pharmacological strategies targeting FKBP51 to modulate autophagy could have differential effects if compounds are isoform-specific. Overall, these results suggest different functional roles or dynamics for the two isoforms. Considering the different functions related to the different domains, it would be of particular interest to explore in more detail the functions related to the unique C-terminal sequence of 46 aa of isoform 2. This could reveal additional effects not shared with isoform 1 that could be amenable to selective drug targeting. While our results show quite distinct functional roles of the two isoforms, the much lower expression of isoform 2 needs to be considered when interpreting overall effects. However, the distinct time dynamic to stimuli, such as activation via glucocorticoids and possibly also other inducers, may open time windows in which isoform 2 is present at substantial levels compared to isoform 1. As mentioned above, the functional role of the distinct subcellular location also remains to be explored, as this could also differentially affect biochemical processes in different cellular compartments.

The identification of functional disparities between FKBP51 isoforms in their dynamic regulation following stimulation carries significant weight for future research on FKBP5/51 and drug design. The central role of FKBP51 as a molecular scaffolding protein interacting with various partners and regulating multiple pathways highlights the importance of selective regulation in the context of targeted therapeutic approaches. We have unveiled a differential dynamic of the two FKBP51 isoforms in the regulation of GR activation. Given the modulatory role of FKBP51 on other steroid receptors, such as estrogen [35], progesterone [36], and androgen receptors [37], we can speculate an analogous effect with relevant consequences for associated diseases and therapies. Furthermore, a deeper understanding of the distinct roles and dynamics of FKBP51 isoforms offers the potential for more precise interventions in stress-related diseases, including metabolic dysfunctions, which are significant comorbidities in depression and arise as side effects of the commonly used psychopharmacological treatments. FKBP51 has been implicated in metabolic disorders through its modulation of the insulin receptor pathway [38,39] and its correlation with leptin signaling [40], as well as by interacting with peroxisome proliferator-activated receptor-γ (PPARγ) [4,41] and regulating the energy sensor AMPK [42]. Furthermore, autophagy—in part differentially regulated by the two FKBP51 isoforms, as we have demonstrated—plays a critical role in catabolic processes [39,43,44]. Genetic modifications of FKBP51 through knockdown or knockout experiments resulted in changes in body weight and had significant effects on autophagy, metabolism, and body weight [31]. Pharmacological interventions using the widely applied small-molecule SAFit/SAFit2 also demonstrated weight-reducing effects in mice [11]. Another important disease-associated role of FKBP51 is in the development of tauopathies, including Alzheimer’s disease [2], and knocking down FKBP51 via antisense oligonucleotides results in reduced tau levels [45]. These studies show that targeting FKBP51 in humans has shown potential to improve FKBP51-associated diseases [45,46], and taking into consideration the differential roles and dynamics of FKBP51 isoforms, could further improve the drug design and, consequently, the outcome for patients. We acknowledge the limitation that most experiments have only been carried out in HeLa cells, so further studies on potential tissue-specific effects will be necessary to increase our knowledge about the medical relevance of isoform-specific drug targeting. The understanding that isoform 1 and isoform 2 not only exhibit different responses to dexamethasone but also have distinct roles in regulating the GR suggests that targeted therapies could be developed to modulate these isoforms selectively and the urgency of future studies to address this difference. Furthermore, in terms of future research on FKBP51, these findings emphasize the need for a comprehensive understanding of the roles of different isoforms in various cellular and subcellular contexts and disease states. While animal models have proven invaluable for understanding FKBP5’s role in both physiological and pathological processes, the absence of isoform 2 in rodents may have created a shortsighted gap in our comprehensive understanding of this scaffold protein. Further investigation into the mechanisms underlying the differential functions of isoform 1 and isoform 2, including their distinct interactions with cellular partners and signaling pathways, is essential. This study sheds light on the functional divergence of FKBP51 isoforms and highlights the importance of including such differentiation in future studies. These findings not only contribute to our understanding of FKBP51 biology but also hold promise for the development of isoform-specific, fine-tuned therapeutic strategies in the treatment of a wide range of diseases.

## 4. Materials and Methods

### 4.1. Antibodies

The following primary antibodies were used for Western blot: BECN1 (1:1000, #3495), ATG12 (1:1000, #2010), LC3B-II/I (1:1000, #2775Cell), AKT (1:1000, #4691), and pAKT (Ser473 1:1000, #4058 and #9275) were acquired from Cell Signaling, Danver, MA, USA; FKBP51 (1:1000, A301-430A, Bethyl, Montgomery, TX, USA, and ab46002, Abcam, Cambridge, UK), FKBP51 specific for isoform 2 (generously provided by the Maria-Fiammetta Romano lab, Federico II University, Naples, Italy); Actin (1:5000, sc-1616, Santa Cruz Biotechnology, Dallas, TX, USA), GAPDH (1:8000, CB1001 Millipore, Burlington, MA, USA).

### 4.2. Plasmids

FKBP51-FLAG as described in Wochnik et al., 2005 [47].

The following expression vectors were purchased from Promega, Madison, WI, USA: FKBP5-pFN21A #FHC02776, GAPDH-pFN21A #FHC02698, HaloTag^®^-pFN21AB8354 #FHC02776, pFN21A HaloTag^®^ CMV Flexi Vector #9PIG282.

HT-FKBP51 isoform 2 expressing plasmid was generated by enzymatic cloning of the coding sequence (ENST00000542713.1) into the pFN21A HaloTag^®^ CMV Flexi Vector.

FKBP51 CRISPR/Cas9 KO Plasmid (h), sc-401560, consisting of a pool of 3 plasmids, each encoding the Cas9 nuclease and a target-specific 20 nt guide RNA (gRNA) designed for maximum knockout efficiency. Of the 3 plasmids, 1 contains gRNA targeting exon 11, specific for variant 1, 2, and 3 but not 4, and the other two plasmids contain gRNAs targeting exon 7 and 4, present in all variants.

### 4.3. RT-qPCR Primers

*FKBP5* variants 1–3 (Exon 11–12), IDT Hs.PT.58.813038: forward primer: AAAAGGCCAAGGAGCACAAC; reverse primer: TTGAGGAGGGGCCGAGTTC.

*FKBP5* all variants (Exon 5–6), IDT Hs.PT.58.20523859: forward primer: GAACCATTTGTCTTTAGTCTTGGC; reverse primer: CGAGGGAATTTTAGGGAGACTG.

*FKBP5* variant 4 (Exon 8–10b), IDT Hs.PT.58.26844122: forward primer: GAGAAGACCACGACATTCCA; reverse primer: AGCCTGCTCCAATTTTTCTTTG.

*YWHAZ* (Exon 9–10), IDT Hs.PT.58.4154200: forward primer: GTCATACAAAGACAGCACGCTA; reverse primer: CCTTCTCCTGCTTCAGCTTC.

### 4.4. Cell Culture

The HeLa cell line (CCL-2, ATCC, Manassas, VA, USA) was cultured at 37 °C, 6% CO_2_ in Dulbecco’s Modified Eagle Medium (Gibco, Fair Lawn, NJ, USA) high glucose with GlutaMAX (31331-028, Thermo Fisher, Fair Lawn, NJ, USA), supplemented with 10% fetal bovine serum (10270-106, Thermo Fisher, Fair Lawn, NJ, USA) and 1% Antibiotic-Antimycotic (15240-062, Thermo Fisher, Fair Lawn, NJ, USA).

The Jurkat cell line (TIB-152, ATCC, Manassas, VA, USA) was cultured at 37 °C, 6% CO_2_ in RPMI (Gibco, Fair Lawn, NJ, USA), supplemented with 10% FCS and 1% Antibiotic-Antimycotic (15240-062, Thermo Fisher, Fair Lawn, NJ, USA).

A final concentration of 100 nM dexamethasone or vehicle (<0.1% DMSO) was used for in vitro treatments for 4 or 24 h in all experiments, except for time and dose curves where times and treatments are specified in the figure legends.

### 4.5. Transfections

Jurkat cells (2 × 10^6^; suspension cells), or with 1× trypsin-EDTA (15400-054, Gibco, Fair Lawn, NJ) detached HeLa cells (2 × 10^6^) were resuspended in 100 μL of transfection buffer [50 mM Hepes (pH 7.3), 90 mM NaCl, 5 mM KCl, and 0.15 mM CaCl_2_]. Up to 2 μg of plasmid DNA was added to the cell suspension, and electroporation was carried out using the Amaxa 2B-Nucleofector system (Lonza, Basel, Switzerland). Cells were replated at a density of 10^5^ cells/cm^2^.

For the intracellular localization experiments, Hela cells were transfected with Lipofectamine 3000 transfection reagent (L3000001, Thermo Fisher, Fair Lawn, NJ, USA), according to the supplier’s protocol.

### 4.6. Imaging

HeLa cells were seeded in 12-well plates and transfected the next day with GFP-control vector, GFP-tagged FKBP51 isoform 1, or GFP-tagged FKBP51 isoform 2. Twenty-four hours after transfection, GFP and brightfield live-imaging were performed using a Zeiss (Jena, Germany) epifluorescence microscope. For GFP imaging, cells were visualized using the GFP filter set (excitation 488 nm, emission 509 nm). Brightfield images were obtained by switching to the transmitted light mode.

### 4.7. Western Blot Analysis

Protein extracts were obtained by lysing cells in 62.5 mM Tris, 2% SDS, and 10% sucrose, supplemented with protease (P2714, Sigma, Setagaya, Japan) and phosphatase (04906837001, Roche, Basel, Switzerland) inhibitor cocktails. Samples were sonicated and heated at 95 °C for 5 min. Proteins were separated by SDS-PAGE and electro-transferred onto nitrocellulose membranes. Blots were placed in Tris-buffered saline solution supplemented with 0.05% Tween (P2287, Sigma-Aldrich Chemie GmbH, Taufkirchen, Germany) (TBS-T) and 5% non-fat milk for 1 h at room temperature and then incubated with primary antibody (diluted in TBS-T) overnight at 4 °C. Subsequently, blots were washed and probed with the respective horseradish–peroxidase- or fluorophore-conjugated secondary antibody for 1 h at room temperature. The immuno-reactive bands were visualized either using ECL detection reagent (WBKL0500, Millipore, Burlington, MA, USA) or directly by excitation of the respective fluorophore. Recording of the band intensities was performed with the ChemiDoc MP system from Bio-Rad, Hercules, CA, USA.

#### Quantification

All protein data were normalized to Actin or GAPDH, which was detected on the same blot. In the case of AKT phosphorylation, the ratio of pAKTS473 to total AKT was calculated. Similarly, the direct ratio of LC3B-II over LC3B-I is also provided, as well as the ratio over Actin.

### 4.8. Real-Time–Quantitative Polymerase Chain Reaction (RT-qPCR)

#### 4.8.1. HeLa Cells

Total RNA was isolated from HeLa cells with the RNeasy mini kit (74104, Qiagen, Hilden, Germany) following the manufacturer’s protocols. Reverse transcription was performed using SuperScript II reverse transcriptase (18064014, Thermo Fisher, Fair Lawn, NJ, USA). Subsequently, the cDNA was amplified in triplicates with the LightCycler 480 Instrument II (Roche, Mannheim, Germany) using primers from IDT and TaqMan^TM^ Fast Advanced Master Mix (4444964, Thermo Fisher, Fair Lawn, NJ, USA).

#### 4.8.2. Human Samples

Participant recruitment, blood withdrawal, and RNA extraction were performed as described by Volk and colleagues [48]. Briefly, 26 males of Caucasian origin aged between 19 and 30 years were recruited (mean age = 25.58 ± 2.64 SD) at the Max Planck Institute of Psychiatry in Munich. All participants were free of a history of psychiatric disorders as well as major neurological and general medical disorders. Further exclusion criteria were regular use of medical drugs, as well as excessive alcohol or caffeine consumption. Unstimulated peripheral blood samples were collected at 12:00 PM, followed by an oral dose of 1.5 mg dexamethasone. Stimulated samples were then taken at 1:00 PM, 3:00 PM, 6:00 PM, and 11:00 AM the next day, corresponding to the 1, 3, 6, and 23 h time points. For RNA extraction, the PAXgene Blood RNA Kit was used, following the QIAGEN protocol for nucleic acid purification. The quality and quantity of the extracted RNA were assessed using the Agilent 2200 TapeStation (Burladingen, Germany), with all samples showing an RNA integrity number (RIN) of ≥7. The 130 samples corresponding to the five time points of the 26 participants were randomized for RT-qPCR to have all the time points for each participant on the same plate and the same assay on the same plate. Samples were run in technical triplicates, and standards were run on each plate to calculate the efficiency. Quality control of the raw data and efficiency-corrected ∆∆CP method published by Pfaffl [49] were performed with R studio (version 2021.09.1, RStudio Team (2021). RStudio: Integrated Development Environment for R. RStudio, PBC, Boston, MA, USA). Statistical analyses were performed with Prism version 9.0.0 (GraphPad Software, La Jolla, CA, USA). Two-way ANOVA and Sidak’s multiple comparisons test were performed.

### 4.9. CRISPR-Cas9 KO Generation

Generation of *FKBP5* KO HeLa cell line: using Lipofectamine 3000 transfection reagent (Thermo Fisher, L3000001), cells were transfected with a pool of three CRISPR/Cas9 plasmids containing gRNA targeting human *FKBP5* and a GFP reporter (sc-401560, Santa Cruz Biotechnology, Dallas, TX, USA). Forty-eight hours post transfection, cells were FACS sorted for GFP as single cells into a 96-well plate using BD FACS ARIA III in FACS medium [PBS, 0.5% BSA Fraction V, 2 mM EDTA, 20 mM Glucose, and 100 U/mL penicillin–streptomycin]. Single clones were expanded, and Western blotting was used to validate the knockouts, and variant-specific knockouts were selected based on Western blot analyses using antibodies specific for Isoform 1 or 2 as detailed above.

### 4.10. Reporter Gene Assays

For the MMTV-luc reporter gene assay, cells were seeded in 96-well plates in medium containing 10% charcoal-stripped, steroid-free serum and cultured for 24 h before transfection using Lipofectamine 2000 as described by the manufacturer. Unless indicated otherwise, the amounts of transfected plasmids per well were 60 ng of steroid-responsive luciferase reporter plasmid MMTV-Luc, 5–7.5 ng of Gaussia-KDEL expression vector as control plasmid, and up to 300 ng of plasmids expressing FKBP51-HTv1; FKBP51-HTv4. If needed, empty expression vector was added to the reaction to equal the total amount of plasmid in all transfections. Twenty-four hours after transfection, cells were cultured in fresh medium supplemented with Dex as indicated or DMSO as control for 24 h. To measure reporter gene activity, cells were washed once with PBS and lysed in 50 µL passive lysis buffer (0.2% Triton X-100, 100 mM K_2_HPO_4_/KH_2_PO_4_ pH 7.8). Firefly and Gaussia luciferase activities were measured in the same aliquot using an automatic luminometer equipped with an injector device (Tristar, Berthold, Bad Wildbad, Germany). Firefly activity was measured first by adding 50 µL Firefly substrate solution (3 mM MgCl_2_, 2.4 mM ATP, 120 µM D-Luciferin) to 10 µL lysate in black microtiter plates. By adding 50 µL Gaussia substrate solution (1.1 M NaCl, 2.2 mM Na_2_EDTA, 0.22 M K_2_H PO^4^/KH_2_PO_4_, pH 5.1, 0.44 mg/mL BSA, Coelenterazine 3 µg/mL), the firefly reaction was quenched and Gaussia luminescence was measured after a 5 s delay. Firefly activity data represent the ratio of background-corrected Firefly to Gaussia luminescence values. The fold stimulation reached saturating concentrations of the hormone at about 3 nM, which is in the range of previous publications [50,51].

### 4.11. Pulse-Chase Assay

Forty-eight hours after transfection with HaloTag^®^-tagged plasmids, cells were labeled with HT fluorescent ligands (HaloTag^®^ R110Direct Ligand, Promega, Madison, WI, USA) for 24 h, after which the fluorescent ligand was washed off (chase) for the indicated amounts of time. Cells were harvested, proteins extracted, minimizing light exposure, and Western blots were performed. Fluorescence was successively measured on membrane with the ChemiDoc MP system from Bio-Rad.

## Figures and Tables

**Figure 2 ijms-25-12318-f002:**
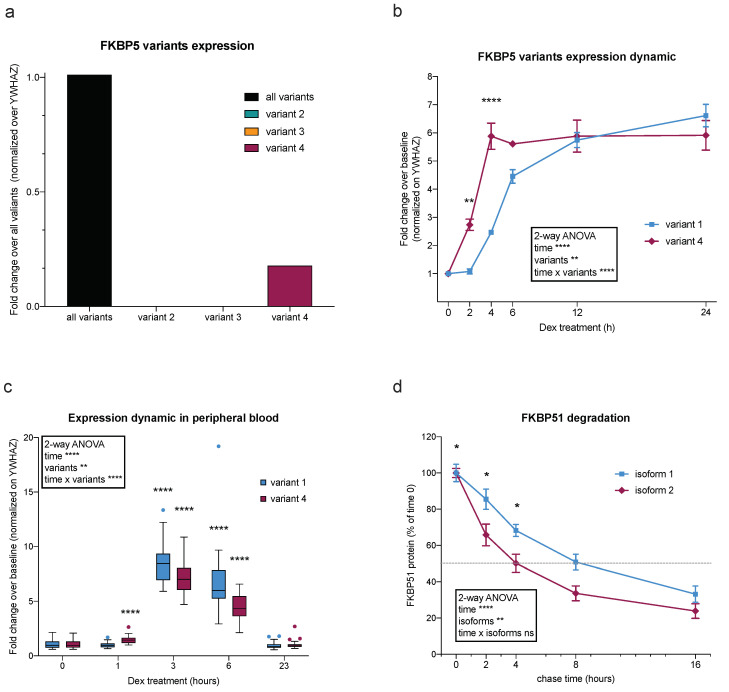
Expression of FKBP5 splicing variants in HeLa cells: (**a**) RT-qPCR quantification of FKBP51 variants in unstimulated HeLa cells. (**b**) RT-qPCR quantification of FKBP51 variants, expressed as fold change in Dex-treated over vehicle-treated, normalized on the housekeeper YWHAZ of HeLa cells treated with 100 nM Dex or vehicle for 24 h. Two-way ANOVA with Geisser–Greenhouse correction (shown in the box) and Sidak’s multiple comparisons test (shown in the graph). Data shown as mean ± SEM. (**c**) Fold change in FKBP5 variants 1 and 4 over vehicle and normalized over YWHAZ at 0, 1, 3, 6, and 23 h after Dex stimulation. Mixed effects model with Geisser–Greenhouse correction (shown in the box) and Sidak’s multiple comparisons test (shown in the graph). Data shown as box-and-whisker plot (Tukey style). (**d**) Pulse-chase assay of FKBP51 isoform 1 and 2 of HeLa cells transfected with HaloTag^®^-tagged-isoform 1 or HaloTag^®^-tagged-isoform 2, pulsed with a fluorophore, and chased for 2, 4, 8, and 16 h. Dotted line indicates half-life of the protein. Quantifications were made from Western blots. * *p* < 0.05. Two-way ANOVA (shown in the box) and Sidak’s multiple comparisons test (shown in the graph). Data shown as mean ± SEM. For all statistics * *p* < 0.05, ** *p* < 0.01, **** *p* < 0.0001, ns = not significant.

**Figure 3 ijms-25-12318-f003:**
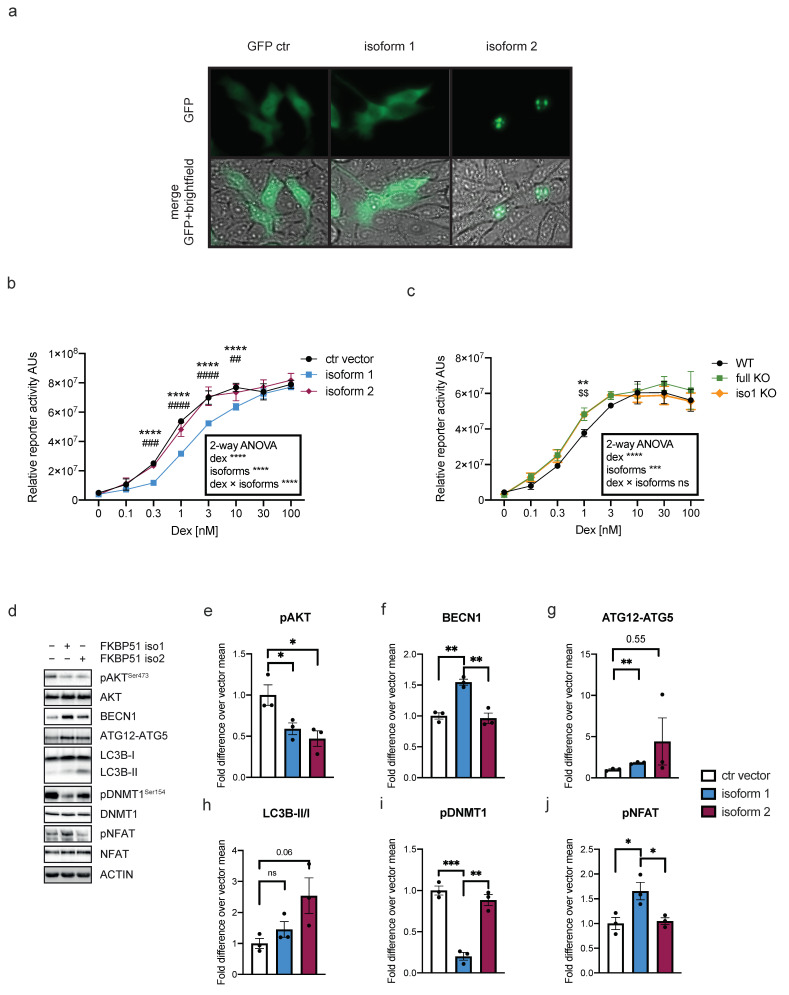
Differential pathway regulation of FKBP51 isoforms: (**a**) Epifluorescent and bright field imaging of HeLa cells transfected with GFP-control vector, GFP-tagged FKBP51 isoform 1 or GFP-tagged FKBP51 isoform 2 24 h prior to imaging. (**b**,**c**) GRE-driven reporter gene assay performed in HeLa cells transfected with (**b**) FKBP51 isoform 1, FKBP51 isoform 2 or an empty vector (ctr vector), or (**c**) in WT (expressing both isoform), full KO and Isoform 1 KO (iso1 KO) HeLa cells treated with 0.1 nM, 0.3 nM, 1 nM, 3 nM, 10 nM, 30 nM, 100 nM, or vehicle for 4 h. Two-way ANOVA (shown in the box) with Tukey multiple comparisons test (shown in the graph). * indicates comparison with control/WT and isoform 1/full KO, # indicates comparison between isoform 1 and isoform 2, and $ refers to comparison between WT and iso 1 KO. (** & ## & $$ *p* < 0.01, *** & ### *p* < 0.0005, **** & #### *p* < 0.0001, ns = not significant). (**d**) Representative Western blots for different pathway markers performed on lysates from HeLa cells transfected with FKBP51 isoform 1, FKBP51 isoform 2 or an empty vector (**e**–**j**) Quantification of Western blots analyses displayed in (**d**): (**e**) phosphorylated AKT (pAKT) normalized on total AKT, (**f**–**h**) autophagy markers, BECN1, ATG12 and LC3B-II/I; (**i**) phosphorylated DNMT (pDNMT) normalized on total DNMT; (**j**) phosphorylated NFAT (pNFAT) normalized on total NFAT from Jurkat cells transfected with FKBP51 isoform 1, FKBP51 isoform 2 or an empty vector; * *p* < 0.05, ** *p* < 0.01, *** *p* < 0.0005, ns = not significant. Mann–Whitney test. Data shown as mean ± SEM.

## Data Availability

Data are contained within the article and Appendix A.

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
