# Peer review of "Differential Dynamics and Roles of FKBP51 Isoforms and Their Implications for Targeted Therapies"

_ijms, 2024, doi:10.3390/ijms252212318_

Round 1
Reviewer 1 Report
Comments and Suggestions for Authors
In the original article "Differential Dynamics and Roles of FKBP51 Isoforms and Their Implications for Targeted Therapies," the authors Silvia Martinelli, Kathrin Hafner, Maik Koedel, Janine Knauer-Arloth, Nils C. Gassen, and Elisabeth B. Binder used in vitro models and peripheral blood cells from a human cohort to demonstrate that the two FKBP5 isoforms are dynamically upregulated following dexamethasone treatment. While the paper presents interesting findings, it requires significant and thorough revision.
-
The authors should revise the abstract in accordance with the International Journal of Molecular Sciences (IJMS) guidelines to ensure clarity and conciseness.
-
The references should be carefully revised and formatted according to the IJMS guidelines.
-
The quality of Figure 1 should be enhanced, and the figure legend should be moved below the figure. Additionally, it is not appropriate to include a figure in the introduction. The data from online databases used in the figure should be presented in a more appropriate section, either integrated into a following figure or described in a dedicated paragraph within the results section.
-
The numbering in paragraph 2 (Results) should be corrected. All subsections (2.2, 2.3.1, 2.3.2, 2.3.3, 2.3.4) should be merged into a single paragraph, as only one figure is presented.
-
Figure 3a should be improved, and the control data should be moved to the beginning for better clarity.
-
The Western blot (WB) analysis should be quantified to provide more robust data.
-
In Figure 3c, why is WT (wild type) used as the control? This should be clarified or justified.
-
The "Materials and Methods" section requires substantial improvement. Paragraph numbering should be introduced for better structure (e.g., 4.1: Cell cultures, 4.2: RT-PCR, 4.3: Transfection, etc.). Additionally, immunofluorescence methods, which are currently missing, should be reported.
-
The phrase "please add" should be removed.
-
Could the authors perform a cell viability assay under the various experimental conditions tested to strengthen their findings?
Author Response
We thank the reviewers for their thoughtful comments and feel that the paper has improved by addressing them. A point by point response to each reviewer can be found below, with responses in italics. Changes to the main manuscript are tracked with the Word tracking tool. The lines are referred to the PDF document.
Comment 1: The authors should revise the abstract in accordance with the International Journal of Molecular Sciences(IJMS) guidelines to ensure clarity and conciseness.
Response 1: Thank you for the suggestion. We have now revised the abstract to include more concrete results and additional methodological detail – see below.
The expression of FKBP5, and its resulting protein FKBP51, is strongly induced by glucocorticoids. Numerous studies have explored their involvement in a plethora of cellular processes and diseases. There is, however, a lack of knowledge on the role of different RNA splicing variants and the two protein isoforms, one missing functional C-terminal domains. In this study we use in vitro models (HeLa and Jurkat cells) as well as peripheral blood cells of a human cohort (N=26 male healthy controls) to show that the two expressed variants are both dynamically upregulated following dexamethasone, with significantly earlier increases (starting 1-2 hours after stimulation of the short isoform both in vitro and in vivo. Protein degradation assays in vitro showed a reduced half-life (4 hours vs. 8 hours) of the shorter isoform. Only the shorter isoform showed a subnuclear cellular localization. The two isoforms also differed in their effects on known down-stream cellular pathways, including glucocorticoid receptor function, macroautophagy, immune activation and DNA methylation regulation. The results shed light on the difference of the two variants and highlight the importance of differential analyses in future studies with implications for targeted drug design.
Comment 2: The references should be carefully revised and formatted according to the IJMS guidelines.
Response 2: Thank you for pointing this out, we have corrected the reference style and reviewed all citations.
Comment 3: The quality of Figure 1 should be enhanced, and the figure legend should be moved below the figure. Additionally, it is not appropriate to include a figure in the introduction. The data from online databases used in the figure should be presented in a more appropriate section, either integrated into a following figure or described in a dedicated paragraph within the results section.
Response 3: Thank you for the feedback. We have enhanced the quality of the image and moved the legend below it. As for the position of the reference to Figure 1 in the text, we agree that it is not very common to include a figure in the introduction, but considering that these are not original data, but rather a visual support to the background knowledge necessary to the understanding of the paper, we feel that this justifies the inclusion of this figure in the Introduction and not the Results section.
Comment 4: The numbering in paragraph 2 (Results) should be corrected. All subsections (2.2, 2.3.1, 2.3.2, 2.3.3, 2.3.4) should be merged into a single paragraph, as only one figure is presented.
Response 4: The subsections were removed and merged into a single paragraph.
Comment 5: Figure 3a should be improved, and the control data should be moved to the beginning for better clarity.
Response 5: Thank you for the suggestion, the control was moved to the left for better clarity.
Comment 6: The Western blot (WB) analysis should be quantified to provide more robust data.
Response 6: We apologize that our Figure legend was not sufficiently clear. Figure 3 d represents representative blots, while panels e-j are indeed the quantifications. We changed the figure legend to make this clearer.
Comment 7: In Figure 3c, why is WT (wild type) used as the control? This should be clarified or justified.
Response 7: WT cells express both isoforms, while iso1 KO express only isoform 2 and full KO do not express any isoform of FKBP51. By comparing the three lines, we are able to pinpoint the role of isoform 2 in the dexamethasone-induced GR response. With this experiment we could indeed confirm that cells expressing isoform 2 alone are not able to evoke the same GR response as WT. We have now clarified this in the figure legend and the discussion.
Comment 8: The "Materials and Methods" section requires substantial improvement. Paragraph numbering should be introduced for better structure (e.g., 4.1: Cell cultures, 4.2: RT-PCR, 4.3: Transfection, etc.). Additionally, immunofluorescence methods, which are currently missing, should be reported.
Response 8: Thank you for the suggestion. We introduced numbered subparagraphs and added more information on the human cohort, treatments as well as statistics.
Regarding the immunofluorescence method, we acknowledge an error in the description that may have caused ambiguity. The cells were not seeded on cover glasses, as previously stated by mistake, but were instead seeded directly onto the plastic of 12-well plates. Live imaging was performed 24 hours after transfection. No fixation, immunohistochemistry or counterstaining with any chemical was conducted. For this reason, we described the process under the title “imaging” rather than immunofluorescence. We have now corrected the error and added further details for clarity. We thank the reviewer for pointing this out and hope the protocol is now clear.
Lines 423-428:
HeLa cells were seeded in 12-well plates and transfected the next day with GFP-control vector, GFP-tagged FKBP51 isoform 1 or GFP-tagged FKBP51 isoform 2. 24 hours after transfection, GFP and brightfield live-imaging were performed using a Zeiss epifluorescence microscope. For GFP imaging, cells were visualized using the GFP filter set (excitation 488 nm, emission 509 nm). Brightfield images were obtained by switching to the transmitted light mode.
Comment 9: The phrase "please add" should be removed.
Response 9: Thank you for catching the mistake. The phrase was removed.
Comment 10: Could the authors perform a cell viability assay under the various experimental conditions tested to strengthen their findings?
Response 10: We agree with the reviewer that cell viability is an important issue to consider. However, at the applied dose (100nM) many papers have shown that dexamethasone does not impact cell viability in different cell types and in HeLa cells not even at much higher doses. Kalinec et al,. 2016 [1] for example show no effects of 24 hour exposure to dexamethasone on MTT-based cell viability assays in HeLa cells at a concentration of 400 µM. Other factor, such as HaloTag expression, are commonly used techniques that are proven to have no significant effect on cell viability [2].
- Kalinec, G.; Thein, P.; Park, C.; Kalinec, F. HEI-OC1 Cells as a Model for Investigating Drug Cytotoxicity. Hear Res 2016, 335, 105–117, doi:10.1016/j.heares.2016.02.019.
- Los, G. V.; Encell, L.P.; McDougall, M.G.; Hartzell, D.D.; Karassina, N.; Zimprich, C.; Wood, M.G.; Learish, R.; Ohana, R.F.; Urh, M.; et al. HaloTag: A Novel Protein Labeling Technology for Cell Imaging and Protein Analysis. ACS Chem Biol 2008, 3, 373–382, doi:10.1021/cb800025k.
Reviewer 2 Report
Comments and Suggestions for Authors
The study and its procedure and intents are clear and well described, however, we would like the authors to clarify a few points:
1 A similar study on FKBP5 refers to the T allele of SNPs, rs1360780 (C/T), associated with increased induction of FKBP5 by glucocorticoids. Their outcomes from dexamethasone/corticotropin-releasing hormone (DEX/CRH) and quantitative real-time PCR analysis of peripheral blood mononuclear cell (PBMC) cDNA samples from 174 and 278 individuals found greater suppression of the stress hormone (cortisol) response to DEX/CRH testing (P=0.0016) in older individuals (>50 years) carrying the T allele compared with older carriers who did not carry the T allele. T carriers showed significant age-related changes in GR and FKBP5 mRNA expression levels in PBMC (P=0.0013 and P=0.00048, respectively). The results suggest that FKBP5 rs1360780 regulates HPA axis reactivity and GR in an age-dependent manner.
2 It follows...This study used in vitro models and peripheral blood cells from a human cohort to demonstrate that both expressed variants are dynamically upregulated following dexamethasone administration. The problem that arises here is crucial, (i) there are no reference lines that allow us to translate from in vitro to in vivo, and (ii), the study lacks fundamental assumptions such as blood donor age and sex. Furthermore, the investigation of the subcellular localization of protein isoforms, their degradation dynamics, and their differential role in known cellular pathways remains somehow vague and clinically of little relevance.
3 I suggest authors stress more about the importance of those two variants in a possible clinical scenario...
Comments on the Quality of English LanguageEnglish is good
Author Response
We thank the reviewers for their thoughtful comments and feel that the paper has improved by addressing them. A point by point response to each reviewer can be found below, with responses in italics. Changes to the main manuscript are tracked with the Word tracking tool. The lines are referred to the PDF document.
Comment 1: A similar study on FKBP5 refers to the T allele of SNPs, rs1360780 (C/T), associated with increased induction of FKBP5 by glucocorticoids. Their outcomes from dexamethasone/corticotropin-releasing hormone (DEX/CRH) and quantitative real-time PCR analysis of peripheral blood mononuclear cell (PBMC) cDNA samples from 174 and 278 individuals found greater suppression of the stress hormone (cortisol) response to DEX/CRH testing (P=0.0016) in older individuals (>50 years) carrying the T allele compared with older carriers who did not carry the T allele. T carriers showed significant age-related changes in GR and FKBP5 mRNA expression levels in PBMC (P=0.0013 and P=0.00048, respectively). The results suggest that FKBP5 rs1360780 regulates HPA axis reactivity and GR in an age-dependent manner.
Comment 2: It follows...This study used in vitro models and peripheral blood cells from a human cohort to demonstrate that both expressed variants are dynamically upregulated following dexamethasone administration. The problem that arises here is crucial, (i) there are no reference lines that allow us to translate from in vitro to in vivo, and (ii), the study lacks fundamental assumptions such as blood donor age and sex. Furthermore, the investigation of the subcellular localization of protein isoforms, their degradation dynamics, and their differential role in known cellular pathways remains somehow vague and clinically of little relevance.
Response 1 and 2: Thank you for highlighting these important points. Below we try to point out relevant information for translation from in vitro to in vivo. We observe similar RNA expression trajectories in cell lines and peripheral blood cells from the in vivo experiment. Given that FKBP5 is expressed in most tissues and its function has been studied in many organ systems and diseases, the focus of our work was to highlight differences in properties of the two main isoforms, but not with a specific tissue focus. This would need to be part of additional, future studies. We have now added this as a limitation to the discussion (lines 349-352).
We agree that important details regarding the human participants were only available through the reference. We have now added information about age (19-30 years), sex (all male), and health status (healthy controls) as well as additional details of the human cohort to the Methods section (line 456-465 as well as line 524-526 and below).
Participant recruitment, blood withdrawal and RNA extraction was performed as de-scribed by Volk and colleagues [1]. Briefly, 26 males of Caucasian origin aged between 19 and 30 years were recruited (mean age = 25.58 +/- 2.64 SD) at the Max Planck Institute of Psychiatry in Munich. All participants were free of a history of psychiatric disorders as well as major neurological and general medical disorders. Further exclusion criteria were regular use of medical drugs, as well as excessive alcohol or caffeine consumption. Un-stimulated peripheral blood samples were collected at 12:00 PM, followed by an oral dose of 1.5 mg dexamethasone. Stimulated samples were then taken at 1:00 PM, 3:00 PM, 6:00 PM, and at 11:00 AM the next day, corresponding to the 1, 3, 6 and 23 hours time-points.
The focus of the study was an in-depth understanding of the function of the two protein isoforms of FKBP51 in order to enable more focused drug target studies. For this is important to know about expression dynamics of the two isoforms, their downstream targets but also their subcellular distribution. We had highlighted this in the discussion (338-340, but now also added the potential clinical relevance of the study to the introduction (line 99-109).
FKBP51 is strongly activated with HPA-axis activation and altered HPA-axis function has been observed with many disorders, including psychiatric, cardiovascular and autoimmune diseases [2]. FKBP51 is also strongly upregulated with synthetic glucocorticoids and these are commonly used in the treatment of many conditions and disorders [3]. Understanding differences in the regulation of FKBP51 isoforms following glucocorticoid exposure could have important clinical implications, especially if altered downstream pathways are different. In addition, FKBP51 is actively explored as a drug target in a number of diseases, including posttraumatic stress disorder, neurological diseases, pain, inflammatory disease and cancers [4–8]. A better understanding of the function of different isoforms is important to optimize the development of FKBP51-targeting drugs.
Comment 3: I suggest authors stress more about the importance of those two variants in a possible clinical scenario...
Response 3: We thank the reviewer for this important suggestion. In addition to adding more information on clinical relevance in the introduction, see response 2 above, we have also added text in the discussion to better highlight the potential clinical relevance of isoform-specific drug targeting. See lines 295-296, 320-322, 333-335 and 338-340.
- Volk, N.; Paul, E.D.; Haramati, S.; Eitan, C.; Fields, B.K.K.; Zwang, R.; Gil, S.; Lowry, C.A.; Chen, A. MicroRNA-19b Associates with Ago2 in the Amygdala Following Chronic Stress and Regulates the Adrenergic Receptor Beta 1. J Neurosci 2014, 34, 15070–15082, doi:10.1523/JNEUROSCI.0855-14.2014.
- DeMorrow, S. Role of the Hypothalamic–Pituitary–Adrenal Axis in Health and Disease. Int J Mol Sci 2018, 19, 986, doi:10.3390/ijms19040986.
- Johnson, D.B.; Lopez, M.J.; Kelley, B. Dexamethasone; 2024;
- Annett, S.; Moore, G.; Robson, T. FK506 Binding Proteins and Inflammation Related Signalling Pathways; Basic Biology, Current Status and Future Prospects for Pharmacological Intervention. Pharmacol Ther 2020, 215, 107623, doi:10.1016/j.pharmthera.2020.107623.
- Bailus, B.J.; Scheeler, S.M.; Simons, J.; Sanchez, M.A.; Tshilenge, K.-T.; Creus-Muncunill, J.; Naphade, S.; Lopez-Ramirez, A.; Zhang, N.; Lakshika Madushani, K.; et al. Modulating FKBP5/FKBP51 and Autophagy Lowers HTT (Huntingtin) Levels. Autophagy 2021, 17, 4119–4140, doi:10.1080/15548627.2021.1904489.
- Lesovaya, E.A.; Chudakova, D.; Baida, G.; Zhidkova, E.M.; Kirsanov, K.I.; Yakubovskaya, M.G.; Budunova, I. V. The Long Winding Road to the Safer Glucocorticoid Receptor (GR) Targeting Therapies. Oncotarget 2022,13, 408–424, doi:10.18632/oncotarget.28191.
- Wedel, S.; Mathoor, P.; Rauh, O.; Heymann, T.; Ciotu, C.I.; Fuhrmann, D.C.; Fischer, M.J.M.; Weigert, A.; de Bruin, N.; Hausch, F.; et al. SAFit2 Reduces Neuroinflammation and Ameliorates Nerve Injury-Induced Neuropathic Pain. J Neuroinflammation 2022, 19, 254, doi:10.1186/s12974-022-02615-7.
- Göver, T.; Slezak, M. Targeting Glucocorticoid Receptor Signaling Pathway for Treatment of Stress-Related Brain Disorders. Pharmacological Reports 2024, doi:10.1007/s43440-024-00654-w.
Round 2
Reviewer 1 Report
Comments and Suggestions for Authors
The authors significantly improved the manuscript, so it is now publishable in ijms.
Comments on the Quality of English LanguageMinor editing of English language required.
Author Response
Thank you very much for your feedback.
Reviewer 2 Report
Comments and Suggestions for Authors
I would suggest adding something more
1 Because of the association of FKBP51 with the glucocorticoid receptor (GR) and peroxisome proliferator-activated receptor-γ (PPARγ), early models of FKBP51 action revolve around the control of metabolism and thus is gaining attention as a meaningful biomarker of metabolic dysfunction.
As clinical evidence psychiatric patients often reveal metabolic disorders and easy to gain weight. Is there any link between FKBP51 and leptin for instance, or any link between FKBP51 and testosterone aromatization process in males? Without this part, the paper appears to have low scientific appeal...
Author Response
Comment 1
I would suggest adding something more
1 Because of the association of FKBP51 with the glucocorticoid receptor (GR) and peroxisome proliferator-activated receptor-γ (PPARγ), early models of FKBP51 action revolve around the control of metabolism and thus is gaining attention as a meaningful biomarker of metabolic dysfunction.
As clinical evidence psychiatric patients often reveal metabolic disorders and easy to gain weight. Is there any link between FKBP51 and leptin for instance, or any link between FKBP51 and testosterone aromatization process in males? Without this part, the paper appears to have low scientific appeal...
Response 1
Thank you for raising this excellent point. We added the following consideration on the role of FKBP5 in metabolic dysfunctions to the discussion (lines 346-365):
“The central role of FKBP51 as a molecular scaffolding protein interacting with various partners and regulating multiple pathways, highlights the importance of selective regulation in the context of targeted therapeutic approaches. A deeper understanding of the distinct roles and dynamics of FKBP51 isoforms offers the potential for more precise interventions in stress-related diseases, including metabolic dysfunctions which are significant comorbidities in depression, and arise as side effects of the commonly used psychopharmacological treatments. FKBP51 has been implicated in metabolic disorders through its modulation of the insulin receptor pathway [1,2] and its correlation with leptin signaling [3], as well as by interacting with peroxisome proliferator-activated receptor-γ (PPARγ) [4,5] and regulating the energy sensor AMPK [6]. Furthermore, autophagy – in part differentially regulated by the two FKBP51 isoforms, as we have demonstrated – plays a critical role in catabolic processes [2,7,8]. Genetic modifications of FKBP51 through knockdown or knockout experiments resulted in changes in body weight and had significant effects on autophagy, metabolism, and body weight [9].Pharmacological interventions using the widely applied small molecule SAFit/SAFit2 also demonstrated weight-reducing effects in mice [10]. Targeting FKBP51 in humans has shown potential to improve metabolic dysfunction [11], and taking into consideration the differential roles and dynamics of FKBP51 isoforms, could further improve the drug design and, consequently, the outcome for patients.”
As this is a molecular study, the clinical relevance is mostly speculative, and a deeper discussion would fall outside the scope of the paper.
To our knowledge there is no link between FKBP5 and testosterone aromatization.
- Balsevich, G.; Häusl, A.S.; Meyer, C.W.; Karamihalev, S.; Feng, X.; Pöhlmann, M.L.; Dournes, C.; Uribe-Marino, A.; Santarelli, S.; Labermaier, C.; et al. Stress-Responsive FKBP51 Regulates AKT2-AS160 Signaling and Metabolic Function. Nat Commun 2017, 8, 1725, doi:10.1038/s41467-017-01783-y.
- Yamamoto, S.; Kuramoto, K.; Wang, N.; Situ, X.; Priyadarshini, M.; Zhang, W.; Cordoba-Chacon, J.; Layden, B.T.; He, C. Autophagy Differentially Regulates Insulin Production and Insulin Sensitivity. Cell Rep2018, 23, 3286–3299, doi:10.1016/j.celrep.2018.05.032.
- Soukas, A.; Cohen, P.; Socci, N.D.; Friedman, J.M. Leptin-Specific Patterns of Gene Expression in White Adipose Tissue. Genes Dev 2000, 14, 963–980, doi:10.1101/gad.14.8.963.
- Häusl, A.S.; Balsevich, G.; Gassen, N.C.; Schmidt, M. V. Focus on FKBP51: A Molecular Link between Stress and Metabolic Disorders. Mol Metab 2019, 29, 170–181, doi:10.1016/j.molmet.2019.09.003.
- Smedlund, K.B.; Sanchez, E.R.; Hinds, T.D. FKBP51 and the Molecular Chaperoning of Metabolism. Trends in Endocrinology & Metabolism 2021, 32, 862–874, doi:10.1016/j.tem.2021.08.003.
- Häusl, A.S.; Bajaj, T.; Brix, L.M.; Pöhlmann, M.L.; Hafner, K.; De Angelis, M.; Nagler, J.; Dethloff, F.; Balsevich, G.; Schramm, K.-W.; et al. Mediobasal Hypothalamic FKBP51 Acts as a Molecular Switch Linking Autophagy to Whole-Body Metabolism. Sci Adv 2022, 8, doi:10.1126/sciadv.abi4797.
- Singh, R.; Kaushik, S.; Wang, Y.; Xiang, Y.; Novak, I.; Komatsu, M.; Tanaka, K.; Cuervo, A.M.; Czaja, M.J. Autophagy Regulates Lipid Metabolism. Nature 2009, 458, 1131–1135, doi:10.1038/nature07976.
- Singh, R. Autophagy in the Control of Food Intake. Adipocyte 2012, 1, 75–79, doi:10.4161/adip.18966.
- Gassen, N.C.; Hartmann, J.; Zschocke, J.; Stepan, J.; Hafner, K.; Zellner, A.; Kirmeier, T.; Kollmannsberger, L.; Wagner, K. V.; Dedic, N.; et al. Association of FKBP51 with Priming of Autophagy Pathways and Mediation of Antidepressant Treatment Response: Evidence in Cells, Mice, and Humans. PLoS Med 2014, 11, e1001755, doi:10.1371/journal.pmed.1001755.
- Hähle, A.; Merz, S.; Meyners, C.; Hausch, F. The Many Faces of FKBP51. Biomolecules 2019, 9, 35, doi:10.3390/biom9010035.
- Hinds, T.D.; John, K.; McBeth, L.; Trabbic, C.J.; Sanchez, E.R. Timcodar (VX-853) Is a Non-FKBP12 Binding Macrolide Derivative That Inhibits PPAR γ and Suppresses Adipogenesis. PPAR Res 2016, 2016, 1–10, doi:10.1155/2016/6218637.
Round 3
Reviewer 2 Report
Comments and Suggestions for Authors
There is just one note to be added
1 It is accepted that the FK1 domains of both immunophilins are the major structural elements responsible for the divergent properties of FKBP51 and FKBP52 on steroid, estrogen, testosterone, and cortisol receptor action. For example, FKBP51 and FKBP52 are also highly expressed in prostate cancer cells. The immunophilin appears to enhance the efficiency of the biological actions of AR, a property of particular relevance for those cases of androgen ablation-based therapies. Several different teams have shown that T treatment combined with E2 induces a prostate cancer incidence of 100% in NBL rats. These results have led to hypothesize that estrogen and aromatization of T to E2 play a critical role in prostate carcinogenesis, at least in the rat (Velasco AM, et al.; Bosland MC, et al: Ozten N, et al.; Dias JP, et al.).
Author Response
Comment 1
There is just one note to be added
1 It is accepted that the FK1 domains of both immunophilins are the major structural elements responsible for the divergent properties of FKBP51 and FKBP52 on steroid, estrogen, testosterone, and cortisol receptor action. For example, FKBP51 and FKBP52 are also highly expressed in prostate cancer cells. The immunophilin appears to enhance the efficiency of the biological actions of AR, a property of particular relevance for those cases of androgen ablation-based therapies. Several different teams have shown that T treatment combined with E2 induces a prostate cancer incidence of 100% in NBL rats. These results have led to hypothesize that estrogen and aromatization of T to E2 play a critical role in prostate carcinogenesis, at least in the rat (Velasco AM, et al.; Bosland MC, et al: Ozten N, et al.; Dias JP, et al.).
Response 1
We thank the reviewer for addressing this relevant perspective. We added the relevance of considering both FKBP51 isoforms in the regulation of androgen receptors as well as other steroid receptors (lines 350-353). For completion, we also added the clinical relevance of FKBP51 isoforms in tauopathies (lines 367-369).
Of note, while the FK1 domain might be relevant for the differential functions of FKBP51 and 52, its function is only secondary to the role of the C-terminal TPR, which is the main domain responsible for the interaction with steroid receptors via HSP-90.
Here the modified part of the discussion with the added parts in yellow:
”The identification of functional disparities between FKBP51 isoforms in their dynamic regulation following stimulation, carries significant weight for future research on FKBP5/51 and for drug design. The central role of FKBP51 as a molecular scaffolding protein interacting with various partners and regulating multiple pathways, highlights the importance of selective regulation in the context of targeted therapeutic approaches. We have unveiled a differential dynamic of the two FKBP51 isoforms in the regulation of GR activation. Given the modulatory role of FKBP51 on other steroid receptors such as estrogen [1], progesterone [2] and androgen receptors [3] we can speculate an analogous effect with relevant consequences for associated diseases and therapies. Furthermore, a deeper understanding of the distinct roles and dynamics of FKBP51 isoforms offers the potential for more precise interventions in stress-related diseases, including metabolic dysfunctions which are significant comorbidities in depression, and arise as side effects of the commonly used psychopharmacological treatments. FKBP51 has been implicated in metabolic disorders through its modulation of the insulin receptor pathway [4,5] and its correlation with leptin signaling [6], as well as by interacting with peroxisome proliferator-activated receptor-γ (PPARγ) [7,8] and regulating the energy sensor AMPK [9]. Furthermore, autophagy – in part differentially regulated by the two FKBP51 isoforms, as we have demonstrated – plays a critical role in catabolic processes [5,10,11]. Genetic modifications of FKBP51 through knockdown or knockout experiments resulted in changes in body weight and had significant effects on autophagy, metabolism, and body weight [12]. Pharmacological interventions using the widely applied small molecule SAFit/SAFit2 also demonstrated weight-reducing effects in mice [13]. Another important disease-associated role of FKBP51 is in the development of tauopathies including Alzheimer’s disease [14], and knocking down FKBP51 via antisense oligonucleotides results in reduced tau levels [15]. These studies show that targeting FKBP51 in humans has shown potential to improve FKBP51-associated diseases [15,16], and taking into consideration the differential roles and dynamics of FKBP51 isoforms, could further improve the drug design and, consequently, the outcome for patients."
- Shrestha, S.; Sun, Y.; Lufkin, T.; Kraus, P.; Or, Y.; Garcia, Y.A.; Guy, N.; Ramos, P.; Cox, M.B.; Tay, F.; et al. Tetratricopeptide Repeat Domain 9A Negatively Regulates Estrogen Receptor Alpha Activity. Int J Biol Sci 2015, 11, 434–447, doi:10.7150/ijbs.9311.
- Barent, R.L.; Nair, S.C.; Carr, D.C.; Ruan, Y.; Rimerman, R.A.; Fulton, J.; Zhang, Y.; Smith, D.F. Analysis of FKBP51/FKBP52 Chimeras and Mutants for Hsp90 Binding and Association with Progesterone Receptor Complexes. Molecular Endocrinology1998, 12, 342–354, doi:10.1210/mend.12.3.0075.
- Stechschulte, L.A.; Sanchez, E.R. FKBP51—a Selective Modulator of Glucocorticoid and Androgen Sensitivity. Curr Opin Pharmacol 2011, 11, 332–337, doi:10.1016/j.coph.2011.04.012.
- Balsevich, G.; Häusl, A.S.; Meyer, C.W.; Karamihalev, S.; Feng, X.; Pöhlmann, M.L.; Dournes, C.; Uribe-Marino, A.; Santarelli, S.; Labermaier, C.; et al. Stress-Responsive FKBP51 Regulates AKT2-AS160 Signaling and Metabolic Function. Nat Commun2017, 8, 1725, doi:10.1038/s41467-017-01783-y.
- Yamamoto, S.; Kuramoto, K.; Wang, N.; Situ, X.; Priyadarshini, M.; Zhang, W.; Cordoba-Chacon, J.; Layden, B.T.; He, C. Autophagy Differentially Regulates Insulin Production and Insulin Sensitivity. Cell Rep 2018, 23, 3286–3299, doi:10.1016/j.celrep.2018.05.032.
- Soukas, A.; Cohen, P.; Socci, N.D.; Friedman, J.M. Leptin-Specific Patterns of Gene Expression in White Adipose Tissue. Genes Dev 2000, 14, 963–980, doi:10.1101/gad.14.8.963.
- Häusl, A.S.; Balsevich, G.; Gassen, N.C.; Schmidt, M. V. Focus on FKBP51: A Molecular Link between Stress and Metabolic Disorders. Mol Metab 2019, 29, 170–181, doi:10.1016/j.molmet.2019.09.003.
- Smedlund, K.B.; Sanchez, E.R.; Hinds, T.D. FKBP51 and the Molecular Chaperoning of Metabolism. Trends in Endocrinology & Metabolism 2021, 32, 862–874, doi:10.1016/j.tem.2021.08.003.
- Häusl, A.S.; Bajaj, T.; Brix, L.M.; Pöhlmann, M.L.; Hafner, K.; De Angelis, M.; Nagler, J.; Dethloff, F.; Balsevich, G.; Schramm, K.-W.; et al. Mediobasal Hypothalamic FKBP51 Acts as a Molecular Switch Linking Autophagy to Whole-Body Metabolism. Sci Adv 2022, 8, doi:10.1126/sciadv.abi4797.
- Singh, R.; Kaushik, S.; Wang, Y.; Xiang, Y.; Novak, I.; Komatsu, M.; Tanaka, K.; Cuervo, A.M.; Czaja, M.J. Autophagy Regulates Lipid Metabolism. Nature 2009, 458, 1131–1135, doi:10.1038/nature07976.
- Singh, R. Autophagy in the Control of Food Intake. Adipocyte 2012, 1, 75–79, doi:10.4161/adip.18966.
- Gassen, N.C.; Hartmann, J.; Zschocke, J.; Stepan, J.; Hafner, K.; Zellner, A.; Kirmeier, T.; Kollmannsberger, L.; Wagner, K. V.; Dedic, N.; et al. Association of FKBP51 with Priming of Autophagy Pathways and Mediation of Antidepressant Treatment Response: Evidence in Cells, Mice, and Humans. PLoS Med 2014, 11, e1001755, doi:10.1371/journal.pmed.1001755.
- Hähle, A.; Merz, S.; Meyners, C.; Hausch, F. The Many Faces of FKBP51. Biomolecules 2019, 9, 35, doi:10.3390/biom9010035.
- Blair, L.J.; Baker, J.D.; Sabbagh, J.J.; Dickey, C.A. The Emerging Role of Peptidyl-Prolyl Isomerase Chaperones in Tau Oligomerization, Amyloid Processing, and Alzheimer’s Disease. J Neurochem 2015, 133, 1–13, doi:10.1111/jnc.13033.
- Gebru, N.T.; Hill, S.E.; Blair, L.J. Development of FKBP5 ASOs to Mitigate Tau Pathology. Alzheimer’s & Dementia 2023, 19, doi:10.1002/alz.082780.
- Hinds, T.D.; John, K.; McBeth, L.; Trabbic, C.J.; Sanchez, E.R. Timcodar (VX-853) Is a Non-FKBP12 Binding Macrolide Derivative That Inhibits PPAR γ and Suppresses Adipogenesis. PPAR Res 2016, 2016, 1–10, doi:10.1155/2016/6218637.